# Challenges That Vhavenḓa Child Victims of Sexual Abuse Endure as a Result of Delayed Disclosure

**Ramphabana Livhuwani Bethuel and Rapholo Selelo Frank \***

Department of Social Work, University of Limpopo, Sovenga 0727, South Africa; ramphabanal@ukzn.ac.za
*   Correspondence: frank.rapholo@ul.ac.za

**Abstract:** While child sexual abuse (CSA) has been declared a public health concern worldwide, the continuous problem is the delayed disclosure that affects child victims in many ways. It poses profound health, psychological and social challenges for child victims. This qualitative study explored and described the challenges that Vhavenḓa child victims of CSA endure as a result of delayed disclosure. Five adult survivors of CSA, seven caregivers of survivors of CSA, four social workers, two educators and two traditional leaders were purposefully selected to participate in the study. Data was collected through semi-structured interviews and thereafter it was arranged and analysed thematically with the assistance of the NVivo software. Findings reveal that, due to the emotional implications of the abuse, the Vhavenḓa child victims of sexual abuse endure a number of behavioural, social and academic challenges that tremendously affect their well-being, even in their adulthood. It is therefore recommended that vigorous awareness campaigns on CSA within the Vhavenḓa communities be conducted to mitigate the delayed disclosure of sexual abuse.

**Keywords:** child victims; child sexual abuse; delayed disclosure; Vhavenḓa

## 1. Introduction

Child sexual abuse is a serious problem and traumatic experience that threatens the human rights and comprehensive development of children worldwide. It brings about short- and long-term negative impacts on the overall development of children, irrespective of their geographical, racial, and physical makeup (Fisher et al. 2017; Ramphabana et al. 2022). It presents adverse physical, mental, behavioural and social impacts on the victims (Alaggia and Millington 2008; Fontes and Plummer 2010; Easton 2014). In addition, Mathews (2019) avers that CSA is widespread across societies, although some report a higher prevalence than others. About 80% of sexual abuse cases remain delayed or unreported to appropriate legal bodies, resulting in a poor understanding of the prevalence of sexual abuse (Alaggia 2010; Kenny and McEachern 2000). Social and cultural contexts are not exempted from influencing the delayed disclosure of CSA. It is difficult to obtain the exact depiction of the prevalence of CSA, due to delayed or lack of disclosure.

Both the delay of disclosure and the complete lack of disclosure make it a hurdle to combat CSA and provide appropriate and effective interventions to victims of CSA. The disclosure of CSA carries a significant potential to provide timeous services and prevent further sexual victimization (Lalor and McElvaney 2010; Alaggia et al. 2019). CSA is a hidden crime that is often perpetuated without witnesses, thus making a disclosure the only way the incident can be understood and accounted for (Alaggia 2004; Hébert et al. 2009; Reitsema and Grietens 2016; Larner 2022; Williams et al. 2022). The disclosure of CSA enjoys comfort in hiding, since sexual abuse often occurs in private, with limited noticeable signs. In other words, providing necessary and timeous services to child victims depends on the disclosure mainly from victims and individuals to whom victims may have confided. It is important to understand what child victims go through, so as to develop appropriate

responses through which the culture of disclosing sexual abuse can be fostered to minimise the short- and long-term impacts of CSA.

Without considering contextual variables, the problem of revealing CSA cannot be fully comprehended. Culture is ideally a system that should contribute significantly to safeguarding and fostering children's holistic development. Children are still victimized in the same cultural surroundings, despite the fact that this is not how it should be. For example, disclosing CSA may be challenging, especially in cultures that place a strong emphasis on maintaining family unity (Alaggia 2001; Paine and Hansen 2002; Ramphabana et al. 2019). The reputation and interests of families are frequently put before those of the abused child, and the effects of the abuse on the child as a victim receive little consideration (Boakye 2009). Ramphabana et al. (2019) also found that socio-cultural practices amongst Vhavenḓa play a significant role both in delayed disclosure and non-disclosure of CSA. For instance, Vhavenḓa children are socialised to conceal information about incidents, including CSA, which could negatively affect the reputation of and unity within families. As a result, child victims are left to suppress their abuse experiences, leaving them with unattended trauma.

The aim of this study was to explore and describe the challenges that Vhavenḓa child victims of CSA endure as a result of delayed disclosure of sexual abuse. Geographically, the Vhavenḓa are a group of people who live in the far northern part of the Limpopo province in South Africa. Beyond the geographical location, the Vhavenḓa have their own culture, language, traditions, beliefs, and practices that distinguish them from other ethnic groups in South Africa. Tabachnick and Pollard (2016) recommended that context-specific research could minimise the potential to overgeneralize the phenomenon of CSA. As much as CSA is a global issue, it is the authors' view that using a 'blanket' or 'one-size-fits-all' approach to understand CSA across different cultural contexts may be insufficient and ineffective, because social norms, beliefs, and practices vary from one context to another. As such, the absence of cultural contextualisation may bring about misinformed recommendations.

## 2. Theoretical Framework

Trauma theory serves as a theoretical framework to provide a basis to look at CSA as a traumatic event that can cause psychological distress and long-term negative effects on the victims. This theory is concerned with how individuals respond to traumatic experiences and how such experiences affect their behaviour, emotions, and thoughts (Becker-Blease and Freyd 2005; Brown et al. 2012; Goodman 2017). Trauma theory suggests that certain events or experiences can overwhelm a person's ability to cope, leading to psychosocial issues and long-term negative effects. CSA is recognised as a particularly severe form of trauma due to its invasive and exploitative nature, especially when it occurs during a child's developmental stages. This theory acknowledges that the effects of traumatic events such as CSA can be enduring, leading to difficulties in trust, intimate relationships, self-esteem, and overall well-being. The theory further argues that delayed disclosure can also negatively affect victims' ability to seek help and support, leading to further isolation and distress. This theory enabled the researchers to explore and describe challenges faced by Vhavenḓa child victims of sexual abuse following the delayed disclosure.

## 3. Materials and Methods

This study adopted a qualitative research approach. Qualitative studies are open-ended and offer researchers opportunities to explore and ascertain the experiences of the people being studied (Holliday 2002; Denzin and Lincoln 2008). The qualitative research approach was ideal and helpful in that it provided the researchers with opportunities to solicit in-depth information regarding challenges faced by Vhavenḓa child victims of CSA in relation to the delayed disclosure of sexual abuse. The researchers employed exploratory and descriptive research designs to achieve the aim of the study. The population of the

study consisted of five adult survivors of CSA, seven caregivers of CSA adult survivors, two traditional leaders, four social workers and two educators in the Vhembe district municipality in Limpopo province who were purposively selected on the basis of the prospect that they would inform the study. The selected population consisted of key stakeholders that are legally expected to report about any form of CSA as stipulated in Section 110 of the Children's Amendment Act 41 of 2007. Semi-structured one-on-one interviews were used as a data collection method. The researchers analysed, arranged, and organised data thematically, with the aid of the NVivo software.

Researchers are responsible for accounting for the quality of the studies they conduct. In qualitative studies, four key criteria (credibility, transferability, dependability, and conformability) are widely used to determine the standard and trustworthiness of a study. To ensure the credibility of the study, prolonged engagements and member checking were used. For transferability, detailed study processes and the methodology followed were provided, so that readers or the audience could decide on the transferability of the findings of the study. For dependability, the researchers provided details on the research design, how data were collected and handled, and coded data correctly. For conformability, the researchers strictly guarded themselves against being clouded by their opinions, attitudes, and personal experiences in reporting the findings. Instead, they reported information collected from the participants.

Strydom (2011) avers that project managers and researchers need to ensure that ethical protocols are duly observed before conducting the study. To that point, permission to conduct the study was obtained from both internal and external structures. At the internal level, ethical clearance was obtained from the Turfloop Research Committee (TREC) at the University of Limpopo, and the project number is TREC/126/2021: PG. At the external level, written permission to conduct the study was obtained from Vhavenḓa traditional leaders, the Department of Social Development (reference number S4/3/2), and the Department of Education (project number LPREC/95/2021: PG) in the Limpopo province.

In terms of the biographical information of participants, four caregivers were within the age category of 31–40, while three were within the age category of 41–50. Three of the adult survivors of CSA who participated in this study were within the age category of 21–30, and two adult survivors were within the age category of 31–40. Four of the five adult survivors were females, while only one was male. Six of the caregivers were females, whereas only one was male. All the two traditional leaders were males; one had traditional leadership experience ranging between 11 and 20 years, while the other had experience ranging between 21 and 30 years. Three of the social workers were females, and had working experience of between 11 and 20 years, while only one was a male, with working experience of between 21 and 30 years. All social workers were registered with the South African Council for Social Services Professions (SACSSP). The two educators were both females, with working experience ranging from 11 to 20 years, and were registered with the South African Council for Educators (SACE).

## 4. Results and Discussion

Guided by trauma theory, the following were found to be the challenges that child victims endure within their families and communities as a result of the delayed disclosure of sexual abuse:

### 4.1. Living with Continued Confusion

Some participants reported that delayed disclosure of CSA makes child victims vulnerable to living with continued confusion. On its own, CSA is a traumatic and confusing experience—and child victims are not immune to any of its impacts. One educator reported that educators are in a better position to identify learners who are abused through learners' conduct in school. On the same note, one of the adult survivors of CSA indicated having a feeling that people do not care about her abuse experience or that such abuse is normal. One social worker reported that CSA victims are sometimes confused when it comes to

whom they should confide in for assistance and protection. Below are responses from the participants:

"Musi ṅwana otambudzwa hayani kana henefho hune angavha otambudzwa hone, u ya kona u muvhona sa mudededzi nga nḓila ine avha a khou ḓifara ngayo musi e kiḽasani. Vhaṅwe zwezwi ri tshi khou vha ṇea mishumo ya tshikolo vhe kiḽasani, uya kona u vha vhona uri huna ndaḓo ine iya kona ulaedza uri u humbulele uri huna zwine ṅwana a khou ṱangana nazwo." (Educator Participant A)

**Translated:** "When a child is abused at home, or wherever the abuse might have happened, you can see as an educator by the way he or she will be behaving in the classroom. For others, when we are giving them some schoolwork or activities in class, you would see that a child is living with confusion and that could lead you to conclude that there is something that the child is facing." (Educator Participant A)

An adult survivor said the following:

"Ndo vha muthu we nda tshila na dzimbudziso nnzhi vhukuma zwi tshi tevhela u tambudzwa hanga lwa vhudzekani. Musi zwi tshi khou dzhia tshifhinga tshilapfu uri muṱa wa hashu u vhige tshiwo itsho, tshiṅwe tshifhinga ndo vha ndi tshi elekana uri kani-ha zwoiteaho khanṇe ndi zwithu zwo doweleaho kana vhathu a vhana ndavha na nṇe. Uvha na dzimbudziso nnzhi zwo vha zwi tshi ita uri ndi tshile ndo ḓaḓa." (Adult Survivor Participant B)

**Translated:** "I lived with a lot of questions following my sexual abuse experience. When it was taking a long time for my family to report the incident, sometimes I would think that maybe what happened to me was usual or people just did not care about me. Having so many questions used to make me live with confusion." (Adult Survivor Participant B)

A social worker said the following:

"Tshiṅwe tshifhinga ṅwana uya tshila o ḓaḓa ngauri u vha usa ḓivhi uri ndi nnyi muthu ane anga pfesesa zwe a ṱangana nazwo khathihi na u vhona uri o tsireledzea." (Social Worker Participant B)

**Translated:** "Sometimes a child would live with confusion because they would not know who would understand what happened to them and ensure that they are protected." (Social Worker Participant B)

The findings above reveal that delayed disclosure of CSA leads child victims to live with continued confusion that may even prolong into their adulthood. These findings are in line with Lev-Wiesel and Daphna-Tekoah (2010) and Jacobs-Kayam and Lev-Wiesel (2019), who found that adult survivors of CSA show high levels of persisting confusion compared to others with no history of CSA. The confusion results from multiple conflicting emotions that a child victim experiences following the abuse. Naturally, parents or caregivers are expected to protect and act in the best interests of children. Therefore, when such expectations are not upheld, child victims are exposed to confusion that prolongs for the long term. It is important to provide Vhavenḓa child victims of CSA with a safe and supportive environment where they feel empowered to disclose their traumatic experiences and receive appropriate interventions.

### 4.2. Bitter Relationship between a Victim and Parent

In general, children depend on their parents for protection and support. Traditionally, parents are responsible for fostering strong and caring relationships with their children (Murovhi et al. 2018). As such, children become resentful and rancorous when parents

seem not keen to care for, protect, and support them. Similarly, when it comes to CSA, child victims would normally believe that their parents would do everything possible to protect them from sexual abuse. Then, when such expectations are not met, a poor parent–child relationship becomes evident. Adlem (2017) also found that it is common for relationship problems between a parent and a child victim to occur following the child's sexual abuse. One adult survivor participant mentioned feeling distanced from her mother, as she felt her mother did not disclose the abuse to the relevant officials. One caregiver participant said the following:

"Phanḓa ha musi ndi tshi vhiga u tambudzwa ha ṅwananga kha vhashumelavhapo, ndo vha ndi tshi zwi vhona uri ro vha si tsheho tsini na tsini. Naho ovha asato zwiamba nga mulomo wawe, ndo vha ndi tshi kona u zwi vhona uri o ntsunyut-shela. Ndo mupfesesa vhunga hu nṋe ane nda fanela u mutsireledza misi yothe"

(Caregiver Participant B)

**Translated:** "Before I reported the sexual abuse case of my daughter to social workers, I could see that there was a distanced relationship between my daughter and I. Although she would not be expressive verbally, I was able to see that she was mad at me. I understood her because I should always protect her." (Caregiver Participant B)

One adult survivor shared the following:

"Nṋe ndo ṱalutshedza mmeanga nga ha u tambudzwa hanga ndi na fhulufhelo ḽa uri vha ḓo ita zwoṱhe uitela uri ndi tsireledzee na uri mupondi wanga a si bvele phanḓa na u ntambudza. Tshiṅwe tshifhinga ndo vha ndi tshi tama uri mupondi amboḓi valelwa kana a tofa zwawe uri ndi si tsha muvhona phanḓa hanga vhunga zwovha zwi tshi nthithisa kha sia ḽa kuhumbulele. Ndo vha ndi tshi pfa uri mmeanga a vho ngo tamba tshipiḓa tshavho nga nḓila yone nahone zwa ita uri ndi sa tshatovha tsinisa navho u fana na phanḓa ha musi tshiwo itshi tshi tshi itea." (Adult Survivor Participant E)

**Translated:** "I told my mother about my abuse experience with the hope that she would do everything to protect me and ensure that the perpetrator would not continue to abuse me. Sometimes I would wish he could be arrested or even die, so that I would not see him anymore, as seeing him used to affect me psychologically. I would feel that my mother did not effectively play her role, and that has created a distance between the two of us, as opposed to how we used to be before the abuse." (Adult Survivor Participant E)

The findings of this study show that delayed disclosure of CSA negatively affects the relationship between parents and child victims. Given the overwhelming psychological, social, and health impacts of CSA, a caring and supporting parent–child relationship becomes important. The above assertion is affirmed by Van Rensburg and Barnard (2005) and Bolen and Lamb (2007) wherein they assert that a caring and supportive parent–child relationship remains crucial in helping child victims cope and heal from sexual abuse. When child victims confide in their parents about their experience of sexual abuse but the parents delay reporting the abuse to the relevant authorities, it poses significant and complex effects on the relationship between the child and their parents. Children could feel betrayed and confused, creating emotional distance between them and their parents. In the context of Vhavenḓa, power dynamics within families could contribute to delaying the disclosure. For instance, parents may be more concerned about the social status of the family than supporting and ensuring that child victims receive the necessary support and intervention. As a result, child victims may doubt their parents' ability to provide a safe and protective environment. It is important to understand that delayed disclosure of CSA

does not only challenge the parent–child relationship, but also hampers and disrupts the child victim's developmental processes.

*4.3. Social Withdrawal*

Some participants reported that the delayed disclosure of CSA affects child victims socially. One adult survivor reported that her parents reprimanded her when she played with other kids, for fear that she would mistakenly disclose the abuse. Willingham (2007) also found that child victims of sexual abuse often withdraw from social activities and find it challenging to interact with peers. One caregiver reported being concerned about seeing her daughter losing interest in playing with others, due to delayed disclosure. Below are some responses from participants:

> "Mmenga vho vha tshi iledza uri ndi ye u tamba na vhaṅwe vhana vha tshi shavha uri ndi nga ḓi wana ndi kha nyimele ine nda ngaita phoswo nda balula zwoiteaho kha nṇe. Hezwo zwoita uri ndi sa tsha ḓiphina nga u tamba na vhaṅwe vhana." (Adult Survivor Participant E)

> **Translated:** "My mother would reprimand me for going to play with other kids, fearing I would be in a situation in which I could mistakenly disclose what happened to me. That has deprived me of the joy of playing with other kids." (Adult Survivor Participant E)

One caregiver said the following:

> "U tambudziwa ha ṅwananga lwa vhudzekani zwoita uri asa tsha tovha na dzangalelo ḽa ubva hayani aya atamba na vhaṅwe. Ndo vha ndi tshi muwana atshi khou mona mona e dzharaṱani na hone atshi khou tamba eeṱhe. Tshiṅwe ndo vha ndi tshi ita ndi tshi humbula uri oṱangana ṱhoho." (Caregiver Participant A)

> **Translated:** "The sexual abuse of my daughter has limited her interest in going out and playing with others. I would find her walking around our yard, playing alone. Sometimes I would even think that she is going crazy." (Caregiver Participant A)

On the same note, one educator said the following:

> "Musi ṅwana o tambadzwa lwa vhudzekani, u ya vhonala nga u sa vha na dzangalelo ḽa u tamba na vhaṅwe musi e tshikoloni. U ya ḓi bvisa kha u vha na vhangana wa wana a tshi ṱoda u vha na tshifhinga eeṱhe." (Educator Participant A)

> **Translated:** "When a child is sexually abused, you could see him or her by a lack of interest in playing with others when s/he is at school. S/he will withdraw from having friends and try to spend time alone." (Educator Participant A)

The findings above show that delayed disclosure of CSA affects how child victims interact with other children. Negative consequences of CSA often bring about widespread challenges with respect to the social interactions of child victims (Victims of Crime 2010; Adlem 2017). Withdrawal from social interactions could be a coping mechanism to suppress feelings of shame, guilt, and pain. In trying to keep the knowledge of the abuse hidden or delaying the disclosure, child victims are sometimes reprimanded from playing with their peers. Unfortunately, this deprives the child victims of the opportunity to play, which could hamper and negatively affect their developmental process, as playing plays a significant role in children's lives and their development. It is important to create a safe and understanding environment where child victims of CSA feel understood, validated and supported, as that could encourage healing and reintegration into social interactions. Following such a traumatic event (CSA), child victims need compassion, empathy, and support.

*4.4. Difficulty Trusting the Opposite Gender*

Some adult survivor participants reported that their sexual abuse experiences affected the manner in which they would look at the opposite gender. One adult survivor indicated that the sexual abuse experience had caused her to develop hate and mistrust of boys and men. On the other hand, a male adult survivor mentioned living with anger towards women, as he was abused by a woman. These findings are resonant with findings by Moelker and Palme (2008) and Adlem (2017) that sexual abuse has the potential to cause sexual problems (including how one views the opposite gender) for the victims. For instance, some victims would engage in homosexual relationships as a result of sexual abuse. Below are some of the responses from the adult survivors of CSA:

"Nṋe zwo ndzhiela tshifhinga tshilapfu uri ndi fhulufhele muthu wa muṱhannga kana munna zwi tshi tevhela u tambudzwa hanga lwa vhudzekani. Ndo vha ndi tshi dzulela u humbula na upfa unga vhaṱhannga na vhanna vhothe ndi vhavhi na hone vha nga kona u vhaisa vhasidzanyana." (Adult Survivor Participant C)

**Translated:** "It took me a long time to trust a boy or man following my sexual abuse experience. I used to think and feel like all boys and men are evil and capable of hurting young girls." (Adult Survivor Participant C)

"Nṋe zwo nkonḓela vhukuma uri ndi ḓi wane ndi tshi vhofholowa musi ndi tshi khou tshila na muthu wa mufumakadzi nga murahu ha utambudzwa hanga lwa vhudzekani. Izwo zwo ita uri ndi sa tsha to vha na dzangalelo la u vha kha vhushaka kana lufuno ngauri zwo vha zwi tshi dzulela uda muhumbuloni wanga uri ndo vhuya nda tambudziwa nga mufumakadzi." (Adult Survivor Participant D)

**Translated:** "I found it difficult to be comfortable living with a woman, due to my sexual abuse. That made me lose interest in having a romantic relationship because it kept coming to my mind that I was once being abused by a woman." (Adult Survivor Participant D)

It can be noted from the findings above that the effects of CSA are not limited to physical implications, but extend to how victims perceive people of the opposite gender as their perpetrators. For instance, if a child victim was abused by a male person, the anger and mistrust would be directed towards males, and vice versa. It is, however, important to highlight that while it is possible for some child victims of sexual abuse to develop anger and mistrust towards individuals of the same gender as their abuser, it is not a universal response. Willingham (2007) shares the fact that CSA has led participants to distrust others, which has made it difficult for them to have heterosexual relationships. With such negative long-term effects, child victims must be assisted in disclosing their abuse experiences so that they may find closure and continue to thrive, even with such atrocious experiences.

*4.5. Poor School Performance*

Some participants reported that CSA affects the academic performance of child victims. This is in corroboration with findings by Crozier and Barth (2005) and Usakli (2012) that when children are exposed to stress they manifest it in different ways, including decreased academic performance. One educator mentioned that child victims of sexual abuse often perform poorly on their academic journey—and others fail several times as a result of the abuse. On the same note, one adult survivor indicated that performing well and focusing on academic work is a challenge following sexual abuse experiences. One of the caregivers also affirmed that the psychological impacts of CSA negatively affect the academic performance of child victims. As such, delayed disclosure could mean that the victims will not receive the necessary support and intervention timeously, which compromises their prospects of succeeding in their academic journey. Below are responses from some participants:

"Ufhedza tshifhinga na vhana tshikoloni zwiita uri ri ṱavhanye uzwi vhona musi huna zwithu zwine azwi khou tshimbila zwavhuḓi kha ṅwana. Vhunzhi ha vhana vho tambudzwaho lwa vhudzekani ri wana kushumele kwavho kha sia ḽa zwa tshikolo ku sato tshimbila zwavhuḓi. Zwiaita na uri vhana avho vhaḓi wane vha tshi khou feila." (Educator Participant B)

**Translated:** "Spending time with children at school makes it easier for us to notice when a child is going through something. For most of the children who are sexually abused, their school performance is often affected negatively, thus making them fail." (Educator Participant B)

One of the adult survivors said the following:

"Nṋe ndo wana zwitshi nkonḓelela uri ndiḓi wane ndi khou futelela kha zwa tshikolo ngauri ndo vha ndi tshi fhedza tshifhinga tshilapfu ndi tshi khou elekanya nga zwe nda ṱangana nazwo. Izwo zwoita uri ndi dovholole mu-role we nda vha ndi khawo nga itsho tshifhinga. Lwovha lwau to thoma ndi tshi dovholola murole kha lwendo lwanga lwoṱhe lwa tshikolo" (Adult Survivor Participant E)

**Translated:** "I found it difficult to focus on the schoolwork because I used to spend a lot of time thinking of what I had gone through. That made me repeat the grade I was doing at the time. It was the first time I repeated a grade in my school journey." (Adult Survivor Participant E)

One of the caregivers said the following:

"U tambudzwa ha ṅwana lwa vhudzekani zwi a tsikeledza ṅwana kha sia ḽa tshikolo vhunga zwi tshi kwama ndiḽa ine a humbula ngayo. Naho ndo vha ndi tshi thusa ṅwananga nga tshuṅwa haya na miṅwe mishumo ya tshikolo, o fhedzisa o feila murole we avha a khawo luvhili lwoṱhe. Naho ovha asatovha muthu ono konesa, murahuni ovha a tshi khwakhwarudzha uswika a tshi phasa." (Caregiver Participant C)

**Translated:** "Child sexual abuse negatively affects a child, as it affects him or her psychologically. Even though I used to help my child with homework and other schoolwork, she ended up failing the grade she was doing twice. Although she was not the top performer before the abuse, she used to try her best to pass." (Caregiver Participant C)

The findings of this study show that delaying the disclosure of CSA brings about negative impacts on the academic performance and development of child victims. Unfortunately, this could cause negative outcomes for the academic and career journey of the child victims (Tillman et al. 2015; Altafim and Linhares 2016). Some child victims may drop out of school as a result of sexual abuse that is not timeously disclosed or reported to the relevant legal authorities for appropriate intervention and support. Some child victims may drop out due to failing continuously, as they find it challenging to deal with their traumatic sexual experiences without professional interventions and support. The impacts of dropout amongst children could increase social ills such as crime, as well as poverty within societies. Ullah et al. (2018) maintain that school dropout among children is one of the social evils that hinders the development of a nation. Thus, it is important to mitigate the non-disclosure of CSA so that child victims can access the necessary interventions and support to continue with their academic journey without distractions.

## 5. Conclusions

CSA is irrefutably a traumatic experience that equally undermines the human rights of children worldwide. The findings of this study show that delayed disclosure of CSA

poses challenges to the comprehensive development of child victims. As such, it is crucial to foster the culture of disclosing CSA as early as possible, so that child victims can access appropriate professional interventions and support, and continue to develop with minimum distractions. The disclosure remains a significant key to accessing services that help child victims to recover from the abuse and minimise the long-term effects of sexual abuse. Furthermore, delaying the disclosure of CSA delays the determination to achieve justice for victims and prevent further abuse. The researchers recommend that awareness campaigns should aim at conscientising Vhavenḓa parents about the dire challenges to which child victims are exposed as a consequence of delaying the disclosure. This should encourage them to always take cognisance of the best interests and human rights of the children. Social workers should work closely with Vhavenḓa traditional leaders (as the custodians of culture and traditions) to help them understand that the quest to encourage disclosure of CSA does not invalidate their culture and traditions, but compliments the efforts to protect children from sexual abuse. Awareness campaigns should help conscientise the Vhavenḓa children about professionals (such as social workers, police officials, and educators) to whom they can report the abuse if their parents seem to be delaying the disclosure. Furthermore, awareness campaigns should help children identify other people to whom they can confide within and outside their immediate surroundings.

**Author Contributions:** Conceptualization, R.L.B. and R.S.F.; methodology, R.L.B. and R.S.F.; software, R.L.B.; validation, R.L.B. and R.S.F.; formal analysis, R.L.B.; investigation, R.L.B.; resources, R.L.B. and R.S.F.; data curation, R.L.B.; writing—original draft preparation, R.L.B.; writing—review and editing, R.S.F.; visualization, R.S.F.; supervision, R.S.F.; project administration, R.L.B.; funding acquisition, n/a. All authors have read and agreed to the published version of the manuscript.

**Funding:** This research received no external funding.

**Institutional Review Board Statement:** The study was conducted in accordance with the Declaration of National Health Research Ethics Council, and approved by the Turfloop Research Ethics Committee (TREC) of the UNIVERSITY OF LIMPOPO (protocol code TREC/01/2018: PG and date of approval: 7 February 2018).

**Informed Consent Statement:** Written consent was obtained from all participants before participating in the study. Given the sensitivity of this study, the researchers have ensured that the data collected are stored in an encrypted drive. For anonymity, the identity details of the participants were substituted with alphabets as per population units.

**Data Availability Statement:** The data presented in this study are available on request from the corresponding author.

**Conflicts of Interest:** The authors declare no conflict of interest.

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
