# Peer review of "Challenges That Vhavenḓa Child Victims of Sexual Abuse Endure as a Result of Delayed Disclosure"

_socsci, doi:10.3390/socsci12070372_

Round 1
Reviewer 1 Report
Dear Author
A specific community was chosen for this study. The motivation why specific this community (Vhavenda) was chosen is not indicated. More information on this community is also needed as it provides the context for the study.
Please provide information on if ethical clearance and legal permission was obtained.
Although it is stated that the study was guided by the trauma theory, this was missing in the discussion of the findings. No link is made between the trauma theory and the findings. A more in-depth discussion of the findings is also necessary.
In the manner that the findings are currently presented, I am also not convinced that the challenges that these participants experienced is due to delayed disclosure and not the trauma of the abuse.
Some minor grammar and technical aspects were noticed.
Reviewer 2 Report
This paper engages with a complex issue, that of delayed disclosure of sexual abuse. The paper unfortunately misses an opportunity to contribute to this area of research and also falls short of its stated intention; to present the challenges that Vhavenda child victims endure as a result of delayed disclosure.
While the focus of the paper is narrow and the data set is small, this is not in and of itself a limitation of the paper. The main limitation is the absence of any context for the study. The issue of disclosure, particularly retrospective studies of same, are well established in the international literature. From the work of Finkelhor in the late 70s and 80s examining the impact of childhood sexual abuse, to more recent work by Ramona Alaggia, Rosaleen McElvaney, Delphine Colin-Vezina and others, charting the impact of disclosure itself, there is a robust evidence-base on this topic - some of which the author's reference. The potential contribution here, is for the authors to examine these impacts and consequences in the specific context of the study - is there something in the cultural and societal context of the Vhavenda population that is different or novel, and can this contribute to the wider knowledge.
The paper unfortunately, does not set out any of the context of the study. It also fails to establish a clear rationale for the study, especially important given the existing work in this area. For example, the aim is to explore challenges faced by child victims but the design only includes adult participants. It is assumed that the study was used a retrospective view point, but this is not clearly articulated or defended.
In terms of the findings posed, many of these relate to more common, and well established, potential impacts of abuse in childhood - issues regarding trust and being believed; hyper-vigilance and concentration issues (impacting education); relationship difficulties related to trust (including relationships with non-offending parents, romantic relationships, and notions of sexuality).
The research design is not clearly articulated in the paper. We are told this was a qualitative design but are not told specifically how was this study conducted, did it receive ethical approval, what was being asked of participants, who conducted the transcriptions? For example, the theme "living with continued confusion" appears to stem from a specific question that may have been posed to participants, as all of the quotes in the paper in this section specifically use this wording 'living with confusion'.
The paper makes some leaps or assumptions that delayed disclosure negatively affects the various issues being discussed (school performance, relationships, even sexual orientation - which is not established). It fails to address the extremely nuanced and complex phenomenon of disclosure and the interaction between personal, inter-personal, and life course issues that play out in individual experiences of abuse and disclosure. The paper therefore provides a limited analysis of these complex inter-relations. An example of this is the somewhat binary assumption, stated in the paper, that experiences of abuse impact an ability to trust the opposite gender.
In terms of recommendations, I would suggest that the author's tie the study specifically to the context - how does the international research relate to the context of Vhavenda communities and to state what, specifically, is the contribution of the paper, what will the reader learn and how is this contributing to the journal and wider knowledge.
I believe the paper could be re-worked, albeit considerably, to present a contribution to the journal.
Round 2
Reviewer 1 Report
Dear authors
Thank you for all the hard work and your efforts to attend to all my concerns that I raised. I am satisfied that all the concerns have been addressed.
Thank you!